# Interferon–Inducible Transmembrane Protein 3 (IFITM3) Restricts Rotavirus Infection

**DOI:** 10.3390/v14112407

**Published:** 2022-10-30

**Authors:** Zhaoxia Pang, Pengfei Hao, Qiaoqiao Qu, Letian Li, Yuhang Jiang, Shuqi Xiao, Ningyi Jin, Chang Li

**Affiliations:** 1College of Veterinary Medicine, Northwest A&F University, Yangling, Xianyang 712100, China; 2Chinese Academy of Medical Sciences, Changchun Institute of Veterinary Medicine, Changchun 130122, China

**Keywords:** IFITM3, rotavirus, cell infection, cell entry

## Abstract

Rotavirus (RV) is a non–enveloped icosahedral virus with an 11–segment double–stranded RNA genome, belonging to the family of rotaviruses. RV is one of the pathogens causing diarrhea in infants and young animals, and it induces the production of type I interferons (IFNs), which can trigger antiviral function by inducing the production of interferon–stimulated genes (ISGs). Although IFITM3, an ISG localizing to late endosomes, can limit many viral infections, whether or not it restricts the infection of RV is still unknown. Therefore, we attempted to determine whether IFITM3 also restricts RV infection by using over–expression and knockout cell strains. It was found that IFITM3–expressing cell strains were less susceptible to RV infection, as the replication of RV in over–expressing cells was significantly less than in control group cells. Correspondingly, IFITM3–knockout cells were significantly susceptible compared to the normal cells. Furthermore, the IFN–induced antiviral effect was significantly attenuated in the absence of IFITM3, and IFITM3 delayed RV escape from endosomes in the presence of IFITM3, suggesting that endogenous IFITM3 is of great importance in type I IFN–mediated antiviral responses and may restrict infection by affecting the function of the late endosomal compartment. In conclusion, these data provide the first evidence that IFITM3 limits RV infection in vitro and delays RV escape from late endosomes into the cytoplasm.

## 1. Introduction

IFITMs play important roles in adaptive and innate immune responses against viruses. Human IFITMs are located on chromosome 11p15.5. IFITM3 is a kind of ISGs, and mainly expresses on endosomes and lysosomes. It has been demonstrated that IFITM3 restricts the infection of a variety of viruses in vitro, such as influenza A virus (IAV), Ebola virus (EBOV), Marburg virus (MAVR), SARS coronavirus (SARS–CoV) [1,2], human immunodeficiency virus (HIV–1) [3], Dengue virus (DENV) [4] and Zika virus (ZIKV) [5]. Regarding the data reported from in vivo experiments, human patients owning non–functional alleles of IFITM3 or mutations in the promoter regions of IFITM3 have also been reported to have an increasing risk of severe IAV infection. Consistent with this, IFITM3 knockout mice have higher sensitivity to IAV and West Nile virus (WNV) [6,7,8], which demonstrates an important role of IFITM3 in antiviral activity. 

IFITM3 is primarily located on early and late endosomal and lysosomal membranes [9]. The molecular mechanisms of IFITM3–mediated antiviral activity have not yet been very clearly investigated, while existing explanations suggest that IFITM3 inhibits viral fusion through a proximity–based mechanism, which inhibits the fusion pore formation by trapping the process at the fusion stage [10,11]. In addition, IFITM3 is considered to directly modify the structure, rigidity and curvature of target membranes, which leads to a block in virus–host fusion [10,12]. As has been reported in other articles, amphotericin B, an antifungal medicine enhancing membrane fluidity, impairs the antiviral activity of IFITM3 [13,14], and IFITM3 protein could be incorporated into nascent virions which lead to decreases in virus entry capacity, such as in HIV–1 virions [15,16,17], EBOV, measles virus (MV), WNV and murine leukemia virus (MLV) [18,19]. Moreover, IFITM3 relies on proper subcellular localization to perform its antiviral functions. Certain sequences at the N–terminal of IFITM3, which is responsible for the localization, are essential for its antiviral activity [6,11]. Furthermore, the palmitoylation of cysteine residues improves the stability of IFITM3 to enhance antiviral ability [20]. 

However, the viruses, mentioned above and restricted by IFITM3, are all enveloped so far. Compared with enveloped viruses, there are fewer studies on non–enveloped viruses, only reoviruses [21], foot–and–mouth disease virus (FMDV) [22] and norovirus (NoV) [23]. Therefore, we conducted a study on whether the IFITM3 molecule affects rotavirus (RV) replication. RV, the cause of severe diarrhea and vomiting in many mammals, can lead to dehydration, electrolyte imbalance and even death [24,25]. RV belongs to the family reoviridae, and it is a non–enveloped icosahedral virus with a ~18,500 bp 11–segment double–stranded RNA genome encoding structure proteins (VP1–4, VP6, VP7) and non–structure proteins (NSP1–NSP6). The six RV structural proteins form three concentric layers [26,27].

It has been reported that ammonium chloride (NH4Cl) and other weak bases prevented the acidification of endosomes and reduced the infectivity of the human strain Wa, while these endosomal acidification inhibitors did not block the infection of the Wa strain, which meant that the Wa strain depended on the acidification of the endosome to infect cells more efficiently [28,29,30]. In addition, some researchers have proposed that the spike protein VP4 undergoes a series of conformational changes to escape from the endosomal compartments into the cytosol to begin replication [31]. Due to IFITM3 localizing on late endosomes as well, we tested whether IFITM3 had an effect on endosomes to restrict RV infection. 

In this study, we found that IFITM3 expression significantly restricted RV infection. IFITM3–mediated restriction mainly occurred at the stage of entry. As the results showed, IFITM3 may limit virus infection in both the early endocytosis and endosomal penetration phases, since virus receptor binding was not affected. This represents the first evidence that IFITM3 can restrict the infection of RV. However, whether IFITM3 delays the dynamics of RV endosomal uncoating leading to inefficient membrane penetration, as well as the difference in the mechanism between enveloped and non—enveloped viruses, require further study.

## 2. Materials and Methods

### 2.1. Cells

Monkey kidney Mark145 cells, Madin–Darby canine kidney (MDCK) cells and Monkey kidney MA104 cells were cultured in Dulbecco’s Modified Eagle Medium (DMEM; HyClone, Logan, UT, USA), containing 10% fetal bovine serum (FBS; Gibco, Grand Island, NY, USA) and 1% penicillin–streptomycin (Solarbio, Beijing, China). Colon carcinoma human Caco2 cells were cultured in the same way, except with 20% FBS. The cells were cultured at 37 °C in a 5% CO2 incubator. Mark145, MDCK and Caco2 cells were kindly stored at Changchun Institute of Veterinary Medicine, Chinese Academy of Agricultural Sciences. 

### 2.2. Viruses

RV human strain Wa and simian strain SA11 were kindly gifted by Professor Wu Yuzhang, Institute of Immunology, PLA, Army Medical University, and they were propagated in MA104 cells as previously described [32,33]. Bat strain MSLH14 is a characterization of a novel G3P [3] RV isolated from a lesser horseshoe bat. It is a distant relative of feline/canine RV, and was propagated in Marc145 cells. RV lysates were activated with trypsin (10 μg/mL; Gibco, Grand Island, NY, USA) for 30 min at 37 °C. All live RV experiments were performed at Biosafety Level 2 at Changchun Institute of Veterinary Medicine, Chinese Academy of Agricultural Sciences.

### 2.3. Virus Titration

Confluent monolayers of MA104 cells or Marc145 in 96–well tissue culture plates were infected with a series of tenfold dilution of trypsin activated RV. RV was diluted in DMEM containing 2% FBS and 1% penicillin–streptomycin. After 5–7 days infection, infected wells were determined by direct examination for cytopathic effect (CPE). Viral titer was measured by 50% tissue culture infective dose (TCID50) by the Reed–Muench method, as previously described [34]. 

### 2.4. RV Infection

RV lysates were activated with 10 μg/mL trypsin for 30 min at 37 °C in advance. Cells at approximately 90% confluence were washed with phosphate–buffered saline and inoculated with RV at a multiplicity of infection (MOI). After 1 h adsorption, the inocula were removed and the cells were maintained in medium containing 2% FBS at 37 °C in a 5% CO2 incubator for an indicated time. Mock–infected cells were generated using culture medium as the control inoculum. Then virus infection was determined by fluorescence imaging, Western blot, flow cytometry, or RT–qPCR.

### 2.5. Over–Expression of IFITM3 in MDCK Cells

The inducible IFITM3–overexpressing MDCK–Tet3G–IFITM3 cells were generated based on the Tet–On 3G system as described previously [35]. Briefly, cells were cotransfected with pTRE3G–IFITM3 and pLVX–Tet3G using Lipofectamine 3000 transfection reagent (Invitrogen, Carlsbad, CA, USA). After 48 h, cells were selected in a complete medium containing 4 μg/mL puromycin (Invivoegn, Carlsbad, CA, USA). Stable cells were harvested and lysed. IFITM3 induced by doxycycline (Dox, 2.5 μg/mL) was examined by RT–qPCR and Western blot. Dox was solubilized in dimethyl sulfoxide (DMSO), so the same volume of DMSO was added into DMEM as the control group.

### 2.6. Knockout of IFITM3 in Caco2 Cells

The IFITM3–konckout Caco2 cells were generated using the CRISPR/Cas9 gene editing system. CRISPR–pLentiv2 vector was stored in the lab. Single guide RNAs were specifically designed using the online tool available at http://crispr.mit.edu (accessed on 20 November 2021). The first sgRNA (sgRNA1) was CACCGGGGGCTGGCCACTGTTGAC (reversed, AAACGTCAACAGTGGCCAGCCCCC). The second sgRNA (sgRNA2) was CACCGTGGATCACGGTGGACGTCGG (reversed, AAACCCGACGTCCACCGTGATCCAC,). Pairs of oligos targeting IFITM3 sequences were annealed and cloned into the BsmBI–digested pLentiv2 vector. Functional lentiviruses were generated in HEK293T cells by cotransfecting IFITM3 sgRNA–encoding CRISPR–pLentiv2, psPAX2 and pMD2.G using Lipofectamine 3000 transfection reagent. HEK293T cells were collected and frozen and thawed twice, then, functional lentiviruses encoding sgRNA1 and sgRNA2 were obtained, respectively. Caco2 cells were coinfected using the two HEK293T cell lysate supernatants for 48 h. After that, 10 μg/mL puromycin was added into culture medium for 24 h, and a single cell clone was generated using limited dilution method in 96–well tissue culture plates. Clones were genotyped and verified by DNA sequencing technique. The sequencing primer was TGCCTGGGCACCATAGTGAAG (reversed, CCAGGGAATGCTCAGAGGGT).

### 2.7. Virus Binding and Entrying Assay

Cell monolayers and virus were pre–chilled on ice for 30 min, mixed at 5 MOI and allowed to bind for 1 h at 4 °C. Under these conditions, the viral particles were allowed to bind on the cell surface, but not to entry into cells. After that, unbound viruses were washed off with ice–cold PBS and then lysed with TRIzol (Life technologies, Carlsbad, CA, USA). Then, the binding virus was assessed by RT–qPCR. After that, the entry process was undertaken by adding pre–warmed growth medium to the cells for the virus entry assay, then the cells were cultured with the medium for 1 h at 37 °C. Finally, cells were washed twice with pre–warmed PBS and total cellular RNA was extracted. 

### 2.8. Endosomal Acidification Inhibition Assay

Cells were grown in 12–well tissue culture plates and adsorbed with human strain Wa at the indicated MOI at 4 °C for 1 h. After adsorption, cells were washed with cold PBS to remove unbound virions, and prewarmed DMEM was added. Then, cells were incubated at 37 °C, and NH4Cl was added at various intervals in order to prevent endosome acidification. Cells were incubated for 12 h and the percentage of infected cells was detected on CytoFLEX (Beckman, Brea, CA, USA).

### 2.9. Analysis of Interferon Sensitivity

Caco2 cells were grownup to 80% confluence in 12–well tissue culture plates. They were then treated with human IFNa2b (5000 U/mL) (Beyotime, Shanghai, China) for 12 h before inoculation with RV at the indicated MOI, and were incubated at 37 °C for a specified time. Finally, the viral RNA load in the cells was assessed by RT–qPCR, and the expression of IFN–induced proteins was detected by Western blot.

### 2.10. Quantitative Real–Time PCR

RNA was purified according to the manufacturer’s instructions (Life technologies, Carlsbad, CA, USA). Total RNA (500 ng) was used as template and was reverse–transcribed to cDNA using M–MLV reverse transcriptase (Promega, Madison, WI, USA). RT–qPCR was performed using SYBR Green Master Mix (TOYOBO, Osaka, Japan) on a BIO–RAD CFX96 Real–Time PCR System (BIO–RAD, Singapore). For relative quantitation analysis, samples were normalized to the expression of housekeeping gene glyceraldehyde–3–phosphate dehydrogenase (GAPDH). The specific primers used were as follows:
**Primer****Sequence (5’****–3’)**RV–NSP5–FTCTATTGGTAGGAGTGAACARV–NSP5–RATGAATCCATAGACACGCCAIFITM3–FATGTCGTCTGGTCCCTGTTCIFITM3–RGTCATGAGGATGCCCAGAATGAPDH–FACCCACTCCTCCACCTTTGACGAPDH–RTGTTGCTGTAGCCAAATTCGTT

### 2.11. Western Blotting

MDCK and Caco2 cells were infected with RV. After infection for an indicated time, the total cell proteins were extracted with IP (Beyotime, Shanghai, China) with 1 mM PMSF (Beyotime, Shanghai, China) on ice for 15 min. The lysates were briefly sonicated, and the concentration of protein in the lysates was assessed with the BCA protein assay reagent (Beyotime, Shanghai, China). The lysates were further denatured by a 10 min incubation in a sample buffer (Phygene, Fuzhou, China) at 100 °C. A certain amount of cell lysates were separated by SDS–polyacrylamide gel electrophoresis and then transferred onto nitrocellulose membranes (Millipore, MA, USA), which were incubated in blocking buffer (5% non–fat milk powder in Tris–buffered saline containing 0.1% Tween 20 (TBS–T)) for 1 h at room temperature. Subsequently, the membrane was incubated in the primary antibody for 16 h at 4 °C. Lastly, the membranes were washed with TBS–T, and then incubated with horseradish peroxidase–conjugated (HRP) antibody for 1 h at room temperature. After a further wash in TBS–T, the enhanced chemiluminescence detection kit (Pierce Biotechnology, Waltham, MA, USA) was used, and the protein bands were detected. Regarding the protein of similar size, such as RV–VP6 (45 kDa) and β–actin (43 kDa), the membrane was washed using stripping buffer (Beyotime, Shanghai, China) to strip previous antibodies, and was then incubated in the next primary antibody. The change in abundance was determined by densitometric analysis. Anti–IFITM3, anti–GAPDH and anti–β–actin antibodies were purchased from Proteintech (Wuhan, China); the anti–RV antibody was purchased from Sata (Santa Cruz, Dallas, TX, USA).

### 2.12. Immunofluorescence 

Cells (4 × 10^4^) were grown in 24—well tissue culture plates and incubated with RV at various MOI at 37 °C for an indicated time. Cells were fixed with 4% paraformaldehyde for 30 min and washed twice with PBS. Then cells were permeabilized with 0.1% Triton X–100, and blocked in PBS with 2% bovine serum. Cells were then incubated with mouse anti–RV antibodies (1:1000) in PBST for 1 h at room temperature and washed twice with PBS–T. Subsequently, cells were incubated with the goat anti–mouse FITC–conjugated antibody (1:1000) (Abcam, Cambridge, UK) in PBST for 1 h at room temperature. Cell nuclei were stained by DAPI (Invitrogen, Carlsbad, CA, USA). Cells were washed twice with PBS. Fluorescence was detected and imaged using a fluorescence microscope (EVOS^TM^ M5000 imaging system; Thermo Fisher Scientific Bothell, WA, USA). Images were analyzed using Adobe Illustrator software. All presented micrographs are representative images of three representative experiments and field images were chosen at random, and the percentage of infected cells was calculated using ImageJ.

### 2.13. Statistics and Reproducibility

Unless otherwise stated, all experiments were repeated at least three times. The data were presented as means ± standard deviation (SD). SPSS 13.0 and GraphPad Prism 9 software were used for statistical analyses. All of the micrographs shown are representative images of three different experiments, as indicated by the legends in the figures. The *p*–values between the two groups for the quantitative data of the indicated experiments were determined using an unpaired *t*–test. Furthermore, for results involving multiple groups, one–way ANOVA was used. *p* values of less than 0.05 were considered significant between the two groups.

## 3. Results

### 3.1. Type I IFN Inhibits RV Infection

The effects of type I IFN on RV strain Wa were investigated by preincubating Caco2 cells with IFNa2b (Biolegend, San Diego, CA, USA) for 12 h. This was followed by RV infection for a certain time. As shown in the RT–qPCR results, Wa infection was significantly limited by IFNa2b (5000 U/mL), and the inhibitory effect of IFNa2b treatment group reached more than eight times that of the mock group at 6 h after viral infection. (Figure 1A). Western blot results also showed that IFNa2b restricted and delayed RV replication, especially in the early stage of RV infection, because the RV–VP6 protein band in the IFNa2b treated group was invisible for the first 6 h (Figure 1B).

### 3.2. Type I IFN and RV Infection Induce the Expression of IFITM3

IFITM3 acts as a type of ISG, as we mentioned above, and the mRNA and protein were examined after IFNa2b (5000 U/mL) treatment on Caco2 cells. As shown in Figure 2A, the RNA transcription of IFITM3 was upregulated after stimulation for 6 h, and was significantly upregulated at 12 and 24 h. Subsequently, IFITM3 protein analysis indicated that it also obviously increased with time before 24 h (Figure 2D). Altogether, these results demonstrated that IFNa2b upregulated IFITM3 expression transcriptionally and translationally. IFITM3 exhibits a broad range of antiviral effects, and is highly expressed in many mammalian cells. In order to analyze whether IFITM3 increased when Caco2 cells were infected by the Wa strain, we tested the mRNA of IFITM3. It showed that Wa–NSP5 significantly increased after infection for 6 h, and reached the maximum at 24 h (Figure 2B). At the same time, IFITM3 significantly increased after infection for 36 h (Figure 2C). It was shown that there was background IFITM3 expression in Caco2 cells, which was further induced after infection (Figure 2E).

### 3.3. Over–Expression of IFITM3 Limits RV Infection 

In order to evaluate the role of IFITM3 in RV infection, MDCK–Tet3G–IFITM3 cells stably over–expressing IFITM3 were previously generated based on the Tet–On 3G system [35]. MDCK–Tet3G–IFITM3 cells were cultured in DMEM for 24 h, DMEM containing Dox or DMSO, respectively. IFITM3 was tested by RT–qPCR and Western blot. Results indicated that DMSO almost had no influence on the mRNA of IFITM3, and DOX significantly stimulated IFITM3 expression (Figure 3A). Western blot showed that IFITM3 was higher in Dox–treated cells than DMSO–treated cells (There is very low background IFITM3 expression in MDCK–Tet3G–IFITM3 cells, to the point of being invisible) (Figure 3I). Thereafter, MDCK–Tet3G–IFITM3 cells were infected with different RV strains at 0.2 MOI for 24 h and 48 h, and VP6 were examined by RT–qPCR. This indicated that MDCK–Tet3G–IFITM3 cells induced by Dox significantly limited the infection of MSLH14 strain, Wa strain and SA11 strain (Figure 3E–G). Next, viral titers were measured by TCID50, and the results showed that there were significant decreases in titers for different strains of RV in Dox–treated MDCK–Tet3G–IFITM3 cells (Figure 3B–D). Furthermore, Wa–VP6 and MSLH14–VP6 were further evaluated by Western blot after infection. A lower VP6 was observed in Dox–treated cells than in DMSO–treated cells (Figure 3H,I). Therefore, these results showed that IFITM3 inhibited infection of different RV strains.

### 3.4. Knockout of IFITM3 Enhances RV Infection

In order to further explore the intrinsic role of IFITM3 upon RV infection in Caco2 cells, CRISPR/Cas9 technology was utilized to generate Caco2–ΔIFITM3 cell strains. After DNA sequencing, a monoclonal cell strain was obtained, and Western blot also validated that the IFITM3 protein was undetectable in Caco2–ΔIFITM3 cells compared with WT cells, even after IFNa2b treatment for 24 h (Figure 4A), suggesting that the IFITM3–knockout cell strain is successfully generated. Next, Caco2–ΔIFITM3 cells were infected with different RV strains in order to further investigate the intrinsic effects of the IFITM3 protein on RV infection. The RV–NSP5 was examined by RT–qPCR. The results indicated that the infection of the Wa strain, SA11 strain and MSLH14 strain in Caco2–ΔIFITM3 cells were enhanced to different degrees in contrast with control cells (Figure 4E–G). Meanwhile, viral titers were measured by TCID50, and results showed that viral titers were significantly increased in Caco2–ΔIFITM3 cells no matter which the strain was (Figure 4B–D). Then, Wa–VP6 was further tested by Western blot after infection for different time. The higher VP6 protein level was observed in Caco2–ΔIFITM3 cells, and the increase was most evident at 6 h (Figure 4H). Overall, these results suggested that the absence of intrinsic IFITM3 protein promoted RV infection. Altogether, all of these results indicated that the absence of IFITM3 promotes RV infection.

### 3.5. Knockout of IFITM3 Attenuates the IFN–Induced Anti–RV Activity 

As previously indicated, IFNa2b remarkably upregulated IFITM3 expression. Meanwhile, the infection of RV was significantly restricted by IFNa2b treatment. Therefore, in order to examine whether IFN–induced anti–RV activity was influenced when IFITM3 was knocked out, RV infection was monitored in Caco2–ΔIFITM3 and Caco2 cells with or without IFNa2b treatment. The Wa strain was chosen to develop the study, and Caco2–ΔIFITM3 and Caco2 cells were treated with IFNa2b (5000 U/mL) for 12 h before cells were infected with the Wa strain. Then, the Wa–NSP5 and Wa–VP6 were, respectively, determined by RT–qPCR and Western blot at 12 h post–infection. As the RT–qPCR results indicated, Caco2–ΔIFITM3 cells were significantly less sensitive to IFNa2b in comparison to Caco2 cells, since the Wa–NSP5 of infected Caco2 cells with IFNa2b pretreatment decreased, on average, by 52 folds, while it only decreased by 2 folds on average in Caco2–ΔIFITM3 cells. The reduction was significantly lower when IFITM3 was absent in cells (*p* < 0.01) (Figure 5A). Wa–VP6 was simultaneously examined. As shown in Figure 5B, the antiviral capacity of IFNa2b was also attenuated, because Wa–VP6 decreased more in Caco2 cells than that in Caco2–ΔIFITM3 cells (2.7–fold versus 1.6–fold) (Figure 5C). Meanwhile, an immunofluorescence assay was taken in order to confirm this phenomenon, and, as the exemplary images indicated, Caco2–ΔIFITM3 cells still showed more severe infection after IFNa2b treatment (Figure 5D). The number of infected cells in two groups was counted using ImageJ, and the results also showed that there was a more dramatic decline in the proportion of infected Caco2 cells than that in Caco2–ΔIFITM3 cells (Figure 5E). Considering all of the above results, it was found that IFNa2b still restricted Wa strain replication in Caco2–ΔIFITM3 cells even if IFITM3 was absent, which indicated that there were some other anti–RV pathways in cells in response to Type I IFN.

### 3.6. IFITM3 Restricts RV Entry and Delays the Dynamics of RV into the Cytoplasm

Next, in order to determine in which stage IFITM3 restricts RV infection, it was hypothesized that IFITM3 would act in the virion disassembly stage on endosomes and not at the stage of receptor recognition and target cell binding. The assay was carried out as follows in Figure 6A. Caco2 or Caco2–ΔIFITM3 cells were adsorbed with the Wa strain at 5 MOI at 4 °C for 1 h, and then binding viruses were detected by Wa–NSP5 using RT–qPCR. Results showed that no differences were observed in infected cells in the presence or absence of IFITM3 (Figure 6B). After binding, cells were incubated for another 1 h at 37 °C, and mRNA of Wa–NSP5 was detected. As results indicated, there were more infections in Caco2–ΔIFITM3 than that in Caco2 cells by roughly two–fold (Figure 6C). 

NH4Cl could restrict Wa strain disassembly by inhibiting the pH decrease required for efficient infection in endosomes [29]. The effects of NH4Cl on the infectivity of the Wa strain in Caco2 cells and Caco2–ΔIFITM3 cells was tested in order to study whether IFITM3 had an effect on Wa release in endosomes. The monolayers of Caco2 and Caco2–ΔIFITM3 cells were adsorbed with Wa at 4 °C for 1 h, and then cells were washed with cold PBS. After adsorption, cells were treated with NH4Cl (0, 20, 40 and 60 mmol/L) in different concentrations and incubated at 37 °C for 24 h in order to determine an appropriate concentration, then they were analyzed by flow cytometry. As results indicated, 20 mM NH4Cl could significantly inhibit Wa infection (Figure 6D), so 20 mM NH4Cl was chosen to carry out the following experiments. 

After adsorption, cells were either mock–treated or treated with NH4Cl (20 mM). NH4Cl was added at various intervals (0, 30, 60 and 120 min) in order to prevent endosome acidification. Then, cells were incubated for 12 h and scored for the percentage of infected cells by flow cytometry (Figure 6E). The results showed that NH4Cl significantly limited Wa infection at early stage, and Wa escaped the NH4Cl blockade at around 120 min in Caco2–ΔIFITM3 cells, while Wa infection was still significantly restricted at the same time in Caco2 cells, which indicated that IFITM3 may delay the kinetics of RV endosomal escape. Given the above results, these data indicate that IFITM3 may restrict RV infection at an early stage. 

## 4. Discussion

The type I IFN system limits viral infection through many mechanisms, and activation of this signaling pathway induces the expression of hundreds of ISGs, which initiate or regulate the immune response and, ultimately, perform antiviral functions [36]. IFITM3 was first identified in 1984 as an ISG, and was a member of the IFITM proteins family [37]. The distribution and expression level of IFITM3 in cells varies depending on cell or tissue type. IFITM3 protein is predominantly expressed in endosomes, and co–localizes with late endosomes and lysosomal markers, such as CD63 and LAMP2, in most cells [37]. Many functional studies have revealed that IFITM3 has broad–spectrum antiviral activities, and has been shown to resist a variety of viruses. Now there is a crucial question of why it has broad–spectrum antiviral properties. Although different viruses enter into the cells in different ways, there are still some similar mechanisms required for viruses to penetrate into host cells. Both enveloped and non–enveloped viruses enter cells through clathrin or caveolae dependent as well as non–caveolae, non–clathrin mediated endocytosis [38]. After endocytosis, enveloped viral penetration involves membrane fusion, and non–enveloped viruses are mediated by pore formation or membrane lysis [39,40,41,42]. Accordingly, some similar pathway requirements of viruses in their entry provide favorable targets for cellular antiviral mechanisms, as certain protein families may resist many potential viruses. We hypothesized that IFITM3 alters RV escape dynamics, possibly by directly or indirectly affecting endosomal function. As expected, the results are also consistent with the prediction that IFITM3 may affect RV in endosomes. A large number of enveloped and non–enveloped viruses enter cells via the endosomal pathway, such as SARS-Cov-2 [43], HIV-1 [44] and Reovirus [21], which probably explains why IFITM3 has broad–spectrum antiviral functions. Therefore, IFITM3 plays a significant role in preventing viral penetration into cells.

Those viruses restricted by IFITM3 protein are more likely to fuse with host cell membranes in endosomes. Previous studies have shown that influenza viruses penetrated into cells through endosomal fusion, and IFITM3 has been widely shown to have an inhibitory effect on influenza viruses. IFITM3 also slowed down the proteolysis of reovirus capsid proteins in endosomes, preventing reovirus releasing to the cytoplasm. In fact, infectious subvirion particles (ISVPs) with equivalent infectivity to reovirus virions, which bypass the endosomal proteases, fuse at the plasma membrane, since there are no differences in the percentage of cells infected by ISVPs regardless of the presence or absence of IFITM3 [21]. When retroviruses, consisting of SARS coronavirus entry proteins, were induced by trypsin to bypass endosomes penetrate cells, IFITM3–mediated resistance was bypassed. However, as an enveloped virus penetrating cells through the endosomal route, IFITM3 does not have effects on adenovirus [1]. In this study, the Wa strain, a non–enveloped virus, was also shown to enter cells through endosomal penetration. Results showed that IFITM3 inhibited RV infection both in Caco2 cells and MDCK cells, despite deriving from different species with different genotypes, including the human–derived Wa strain, simian strain SA11 and bat strain MSLH14. In summary, it is not definite that IFITM3 restricts viruses through endosomal penetration, and thus, how these unaffected viruses escape the affection of IFITM3 needs further study, which may be helpful in explaining the accurate mechanism. In addition, since the RV genome includes 11 segments, it is more likely to have a large number of cross–recombination variants, which also has been proven [45]. Hence, whether IFITM3 resists other RV variants is unclear, and more studies are necessary in the future.

## 5. Conclusions

In conclusion, it was demonstrated that IFITM3 restricted RV infection for the first time and inhibited the cell entry of the human strain Wa. Additionally, IFITM3 possibly functioned in endosomes to delay the dynamics of endosomal escape to cytoplasm. However, data on the more precise antiviral mechanism of IFITM3 is still a long way off.

## Figures and Tables

**Figure 1 viruses-14-02407-f001:**
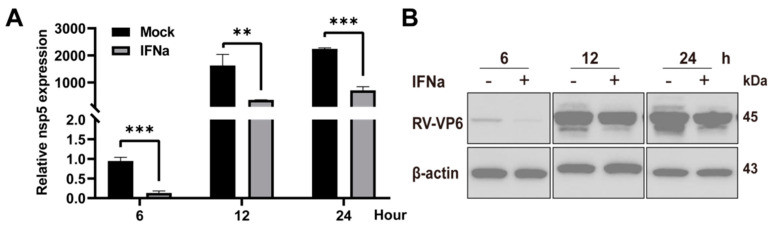
IFNa2b restricts RV infection. Caco2 cells were preincubated with IFNa2b for 12 h and subsequently infected with the Wa strain at 0.2 MOI. (**A**) RT–qPCR analysis of Wa–NSP5 of Caco2 cells infected with the Wa strain at 0.2 MOI for a certain time. Results represented the mean ± SD and were measured in technical duplicates, *n* = 3. **, *p* < 0.01; ***, *p* < 0.001 (Student’s *t* test). (**B**) Western blot analysis of Wa–VP6 in cell lysates at 6, 12 and 24 h post infection.

**Figure 2 viruses-14-02407-f002:**
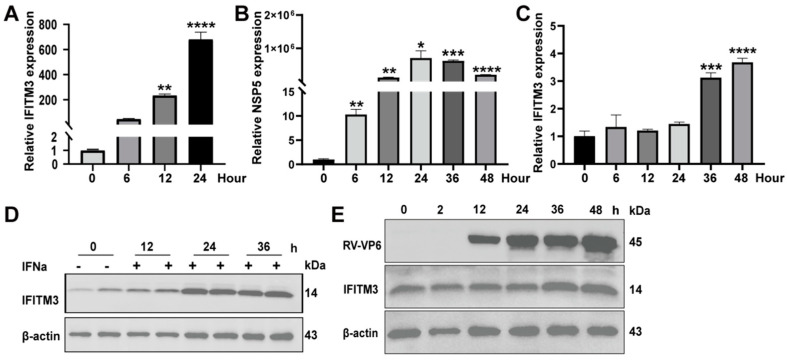
IFNa2b and RV promote the expression of IFITM3. (**A**) RT–qPCR analysis of IFITM3 in Caco2 cells after being incubated with IFNa2b (5000 U/mL) for the indicated time. (**B**) RT–qPCR analysis of the strain Wa–NSP5 in Caco2 cells infected with the strain Wa at 0.2 MOI for the indicated time. (**C**) RT–qPCR analysis of IFITM3 in Caco2 cells infected with the strain Wa at 0.2 MOI for the indicated time. (**D**) Western blot analysis of IFITM3 in Caco2 cells after being incubated with IFNa2b (5000 U/mL) for the indicated time. (**E**) Western blot analysis of the strain Wa–VP6 and IFITM3 in Caco2 cells infected with the Wa strain at 0.2 MOI for certain time (**A**–**C**) Results represented the mean ± SD and were measured in technical duplicates, *n* = 3. *, *p* < 0.05; **, *p* < 0.01; ***, *p* < 0.001 and ****, *p* < 0.0001 (Student’s *t* test).

**Figure 3 viruses-14-02407-f003:**
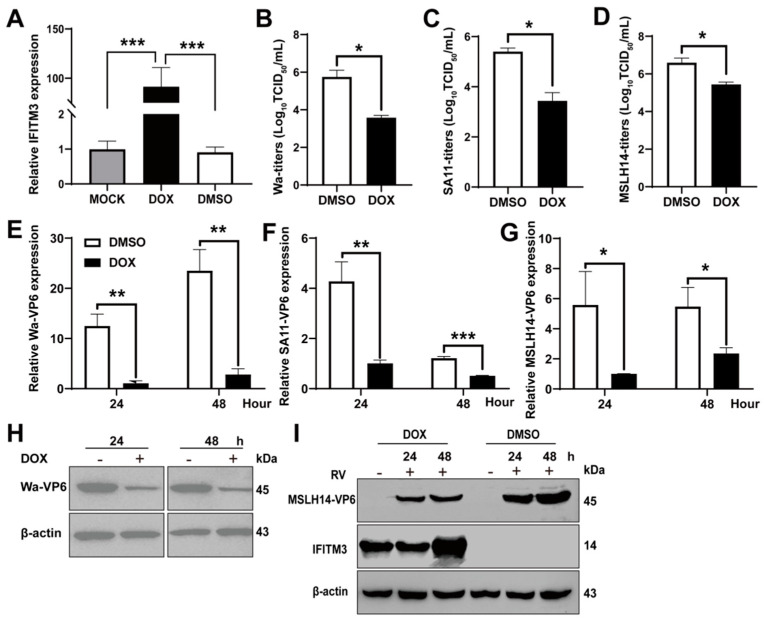
Over–expression of IFITM3 inhibits RV replication in MDCK–Tet3G–IFITM3 cells. (**A**) RT–qPCR analysis of IFITM3 in MDCK–Tet3G–IFITM3 cells after DOX and DMSO treatment. (**B**–**D**) Infectious progeny viral titers in MDCK–Tet3G–IFITM3 cells with or without DOX treatment. The viral titers of the Wa strain, SA11 strain and MSLH14 strain in cells were determined by TCID50. (**E**–**G**) RT–qPCR analysis of VP6 in MDCK–Tet3G–IFITM3 cells infected with the Wa strain, SA11 strain and MSLH14 strain, respectively, at 0.2 MOI for the indicated time. (**H**) Western blot analysis of Wa–VP6 in MDCK–Tet3G–IFITM3 cells after DOX and DMSO treatment. (**I**) Western blot analysis of MSLH14–VP6 in MDCK–Tet3G–IFITM3 cells after DOX and DMSO treatment. (**A**–**G**) Results represented the mean ± SD and were measured in technical duplicates, *n* = 3. *, *p* < 0.05; **, *p* < 0.01; ***, *p* <0.001 (Student’s *t* test).

**Figure 4 viruses-14-02407-f004:**
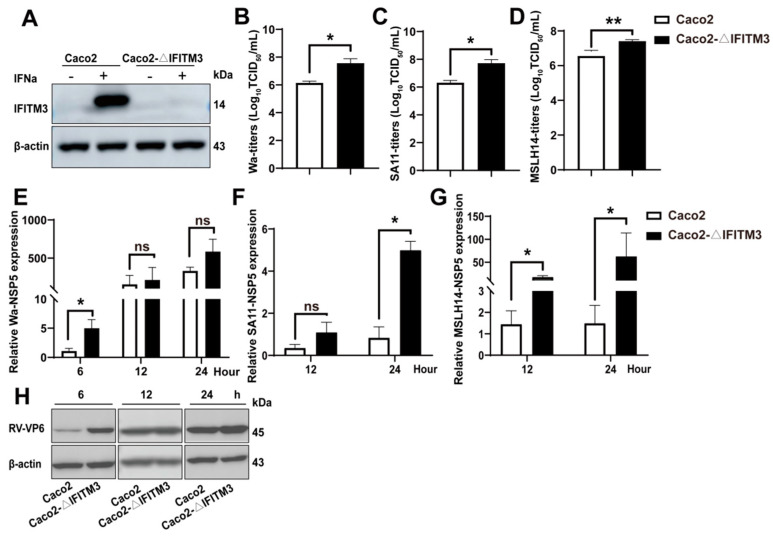
Knockout of IFITM3 enhances RV replication in Caco2 cells. (**A**) Western blot analysis of IFITM3 in Caco2 and Caco2–ΔIFITM3 cells with or without IFNa2b pre–treatment. (**B**–**D**) Infectious progeny viral titers in Caco2 and Caco2–ΔIFITM3 cells. The viral titers of the Wa strain, SA11 strain and MSLH14 strain were determined by TCID50. (**E**–**G**) RT–qPCR analysis of Wa–NSP5 in Caco2 and Caco2–ΔIFITM3 cells infected with the Wa strain, SA11 strain and MSLH14 strain, respectively, at 0.2 MOI for the indicated time. (**H**) Western blot analysis of Wa–VP6 in Caco2 and Caco2–ΔIFITM3 cells after being infected with the Wa strain. (**B**–**G**) Results represented the mean ± SD and were measured in technical duplicates, *n* = 3. ns, *p >* 0.05; *, *p* < 0.05; **, *p* < 0.01 (Student’s *t* test).

**Figure 5 viruses-14-02407-f005:**
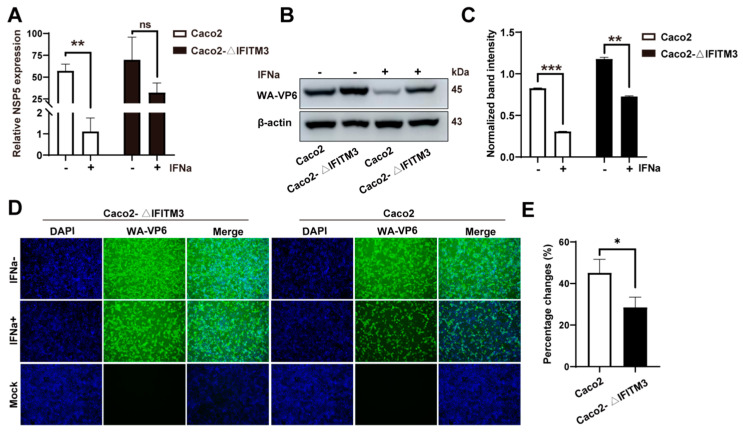
Knockout of IFITM3 attenuates the IFN–induced anti–RV activity. (**A**) RT–qPCR analysis of Wa–NSP5 in Caco2 and Caco2–ΔIFITM3 cells that were, respectively, infected with the Wa strain at 0.2 MOI for 24 h after IFNa2b treatment for 12 h. (**B**) Western blot analysis of Wa–VP6 in Caco2 and Caco2–ΔIFITM3 cells infected with the Wa strain, respectively, at 0.2 MOI for 24 h after IFNa2b treatment for 12 h. (**C**) Bands analysis from panel B were quantified using ImageJ software and normalized to the reference genes β–actin. (**D**) Immunofluorescence analysis (40×) of Wa strain infection for 24 h in Caco2 and Caco2–ΔIFITM3 cells after IFNa2b treatment for 12 h. (**E**) Declined percentages of infected cells in Caco2 and Caco2–ΔIFITM3 cells after IFNa2b treatment. (**A**,**C**,**E**) Results represented the mean ± SD and were measured in technical duplicates, *n* = 3. ns, *p >* 0.05; *, *p* < 0.05; **, *p* < 0.01; ***, *p* <0.001 (Student’s *t* test).

**Figure 6 viruses-14-02407-f006:**
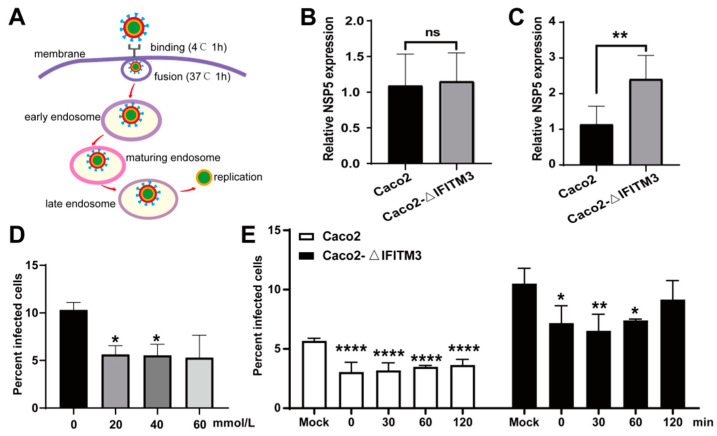
IFITM3 inhibits RV entry and delays Wa strain release from endosomes to cytoplasm in Caco2 cells. (**A**) The model of classical endocytic pathway of RV. Cell entry of the Wa strain began with the binding with glycans on the cell surface. After fusion with the membrane, RV was delivered from early endosomes to maturing endosomes, and finally reached late endosomes that provided the environment to enter the cytosol [28]. (**B**) RT–qPCR analysis of Wa–NSP5 in Caco2 and Caco2–ΔIFITM3 cells that were infected with Wa strain, respectively, at 5 MOI at 4 °C for 1 h. (**C**) RT–qPCR analysis of Wa–NSP5 in Caco2 and Caco2–ΔIFITM3 cells that were incubated, respectively, at 37 °C for a further 1 h after binding. (**D**) Flow cytometry analysis of the percentage of infected Caco2 cells after NH4Cl treatment in different concentrations. (**E**) Flow cytometry analysis of the percentage of infected Caco2 and Caco2–ΔIFITM3 cells after 20 mM NH4Cl treatment. Results represented the mean ± SD and were measured in technical duplicates, *n* = 3. ns, *p >* 0.05; *, *p* < 0.05; **, *p* < 0.01; ****, *p* < 0.001 (Student’s *t* test).

## Data Availability

The raw data supporting the conclusions of this article will be made available by the authors upon request, with consideration of the participants’ privacy and ethical rights.

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
