# Peer review of "Interferon–Inducible Transmembrane Protein 3 (IFITM3) Restricts Rotavirus Infection"

_viruses, 2022, doi:10.3390/v14112407_

Round 1

Reviewer 1 Report

Pang and colleagues provide the first direct evidence that IFITM3 displays antiviral activity against both human and bat-tropic rotavirus strains.  Further, this study suggests that IFITM3 plays the primary role in the antiviral effect of the type-I interferon response.  The authors support their findings with DOX-inducible overexpression of IFITM3, as well as endogenous knockout of the ifitm3 gene.  Differences in rotavirus infectivity were carefully probed using well-controlled qPCR and immunoblotting assays.  Overall, this work is an important contribution to the field of IFITM biology and antiviral research.

Upon reviewing this manuscript, I had a few major concerns that I feel must be addressed to improve the clarity and novelty of the manuscript:

1.  Figures and Figure legends do not match for at least Figures 1, 2, and 3.  Please revise.

2.  Please provide the sgRNA sequences used to target ifitm3 in the methods.

3.  Did the authors evaluate the specificity of their ifitm3 knockout?  It would be important to verify that IFITM2 and IFITM1 are still at expected levels before making specific conclusions about IFITM3. 

4.  Line 242:  IFITM3 expression may be slightly increased after 36h, but to say it is significantly increased requires at least densitometric analysis of at least three western blots.  Please provide this data or reinterpret these results.

5. Figure 3:  What is the effect of DOX treatment on RV infection of MDCK cells?  Is it confirmed that DOX does not indirectly influence RV infection regardless of IFITM3 expression?

6.  Figure 3A:  It is unclear what the "MOCK" condition refers to.

7.  Line 319-322:  I’m not sure the data presented in 5A supports the statement that ifitm3 KO cells are “significantly less sensitive to Type 1 IFN treatment in comparison to control cells.”  To prove this statement true, the authors might consider evaluating the fold IFN inhibition of NSP5 expression in control and knockout cells, then testing for significant differences between the mean fold IFN inhibition of either condition.

8.  Figure 5B:  The IFNa indicators appear to be incorrect.  I assume lanes 1 and 2 are IFN- and lanes 3 and 4 are IFN+?

9.  Figure 5D:  There is no indication of how field images were determined.  Were field images chosen at random?  Are these images representative of multiple similar images?  A quantification of Green/Blue colocalization or mean fluorescence activity of WA-VP6 across multiple random field images would be a more ideal analysis. 

10.  Figure 6:  I am struggling with the interpretation of the NH4Cl treatment experiment for 6E.  It is unclear if this experiment also follows the assay depicted in 6A.  It is also unclear what the Mock condition is referring to, how long the infection in the Mock condition was carried out for, or why there is greater infection in the ifitm3 KO cells compared to control at 0 min if the starting amount of virus is meant to be comparable (as indicated in 6B).  The statistical interpretation of differences between Mock and each time point seems inappropriate, as the ideal comparison would be Mock-treated cells infected at 0, 30, 60, and 120 min compared to NH4Cl-treated cells at the same time points.  As presented, I feel the statistical analyses are skewed by the nearly double percent infection difference between the two conditions, and I can not agree with the conclusion that the effect observed is IFITM3 specific.

Have the authors considered utilizing known inhibitors of IFITM3 to test their hypothesis?  Pretreatment with amphotericin B or rapamycin would presumably increase Caco2 infection by RV, while having much less effect on Caco2 ifitm3 KO cells.

Minor concerns to improve the manuscript:

1.  For all immunoblots, please include the molecular weight in kDa for all bands of interest.

2.  Typo line 316:  “FNa2b”

3.  Line 321:  The reported p value of < 0.05 in the narrative is reported as < 0.01 in the figure legend (Figure 5A).  Also, was significance measured for -/+ IFNa in the ifitm3 KO cells?

4.  Incomplete sentence in line 349.

5.  Line 397:  citation needed

Author Response

Dear Reviewer,

We would like to express our appreciation to you for the helpful suggesting in improving our manuscript entitled “Interferon-inducible Transmembrane Protein 3 (IFITM3) Restricts Rotavirus infection”(Viruses-1940856). We have carefully considered the comments and revised the manuscript according to the comments. With these improvements, we hope that the current version can meet your concern. All revisions to the manuscript were marked up using the MS “Track Changes” function and the version with track changes was resubmitted. The following is a point-by-point responses to your comments and a list of changes we have made to the manuscript.

 Thank you again for your special efforts.

With best regards,

Yours sincerely,

Prof. Chang Li Ph.D.

Changchun Institute of Veterinary Medicine, Chinese Academy of Agricultural Sciences, Changchun, Jilin Province, China;

E-mail: lichang78@163.com

Reviewer 2 Report

Manuscript viruses-1940856 by Pang et al. describes the functional role of IFITM3 in restricting rotavirus infection. Authors describe anti-viral activity of type I IFN against rotavirus and its ability to up-regulate IFITM3. The authors then go on to further prove direct role of IFITM3 in viral inhibition and knockout of this gene enhances viral infection on titer, transcription, and translation level, regardless of rotavirus strains. The authors also prove that IFITM3 partially explains the anti-viral activity of type I IFN like via delaying endosomal escape. 

The manuscript is well written and describe all the procedures towards the functional role of IFITM3 in viral inhibition that authors try to prove.

Some minor comments:

It is recommended that authors review this submission for grammatical accuracy, such as mismatching plural and singular forms of words and confusing verb and noun. For example, on lines 54-55, "IFITM3 protein could be incorporated into nascent virions which leads to decrease virus entry capacity" should be "IFITM3 protein could be incorporated into nascent virions which lead to decrease in virus entry capacity".

It is recommended that authors introduce acronym and stick to using it through the body of the paper. For example, on line98, acronym "RV" is used for rotavirus, then the rest of paper should stick to it. 

It is recommended that authors add word "strain" in front of Wa, SA11 and MSLH14 to clarify rotavirus strain and be consistent. 

It is recommended that authors add descriptor in brackets for fluorescence microscopy, and image system on line206 and for ImageJ software on line335.

Line229: should it be western blot analysis of Wa-VP6 instead of IFITM3

Lines235-236, should refer "Figure 2D" immediately followed by "it increased with time"

Figure 2: please align the labels of panels in legend with those in the figure

Line260: both Figure 3A and Figure 3I are western blot analysis

Figure 3I: please indicate 24 and 48 hour under lanes indicating post infection for both DOX and DMSO treatments.

Section 3.3: it might worth explaining why there is no baseline level of IFITM3 observed under DMSO treatment in MDCK cells.

Lines352-353: should both Figure 6D and Figure 6E to be referred in order to reach the conclusion

Author Response

(The authors gave the same response as above.)
